# Urban-rural differences in hypertension prevalence in low-income and middle-income countries, 1990–2020: A systematic review and meta-analysis

Otavio T. Ranzani[1,2], Anjani Kalra[1], Chiara Di Girolamo[3], Ariadna Curto[1], Fernanda Valerio[4], Jaana I. Halonen[5], Xavier Basagaña[1], Cathryn Tonne[1]*

1 Barcelona Institute for Global Health, ISGlobal, Universitat Pompeu Fabra, CIBER Epidemiología y Salud Pública, Barcelona, Spain, 2 Pulmonary Division, Heart Institute (InCor), Hospital das Clinicas HCFMUSP, Faculdade de Medicina, Universidade de Sao Paulo, São Paulo, Brazil, 3 Health and Social Care Agency, Emilia-Romagna Region, Bologna, Italy, 4 Division of Neuropathology, National Hospital for Neurology and Neurosurgery, Queen Square, London, United Kingdom, 5 Department of Health Security, Finnish Institute for Health and Welfare, Helsinki, Finland

* cathryn.tonne@isglobal.org

**Data Availability Statement:** All relevant data are within the manuscript and its Supporting information files.

## Abstract

### Background

The influence of urbanicity on hypertension prevalence remains poorly understood. We conducted a systematic review and meta-analysis to assess the difference in hypertension prevalence between urban and rural areas in low-income and middle-income countries (LMICs), where the most pronounced urbanisation is underway.

### Methods and findings

We searched PubMed, Web of Science, Scopus, and Embase, from 01/01/1990 to 10/03/2022. We included population-based studies with ≥400 participants 15 years and older, selected by using a valid sampling technique, from LMICs that reported the urban-rural difference in hypertension prevalence using similar blood pressure measurements. We excluded abstracts, reviews, non-English studies, and those with exclusively self-reported hypertension prevalence. Study selection, quality assessment, and data extraction were performed by 2 independent reviewers following a standardised protocol. Our primary outcome was the urban minus rural prevalence of hypertension. Hypertension was defined as systolic blood pressure ≥140 mm Hg and/or diastolic blood pressure as ≥90 mm Hg and could include use of antihypertensive medication, self-reported diagnosis, or both. We investigated heterogeneity using study-level and socioeconomic country-level indicators. We conducted meta-analysis and meta-regression using random-effects models. This systematic review and meta-analysis has been registered with PROSPERO (CRD42018091671).

We included 299 surveys from 66 LMICs, including 19,770,946 participants (mean age 45.4 ± SD = 9 years, 53.0% females and 63.1% from rural areas). The pooled prevalence of

**Funding:** OTR is funded by a Sara Borrell grant from the Instituto de Salud Carlos III (CD19/00110, www.isciii.es). C.T. is funded through a Ramón y Cajal fellowship (RYC-2015–17402, https://www.mineco.gob.es/portal/site/mineco) awarded by the Spanish Ministry of Economy and Competitiveness. OTR, AC, XB and CT acknowledge support from the Spanish State Research Agency and Ministry of Science and Innovation through the Centro de Excelencia Severo Ochoa 2019-2023 Program (CEX2018-000806-S, https://www.ciencia.gob.es/site-web/en/Organismos-y-Centros/Centros-y-Unidades-de-Excelencia.html), and support from the Generalitat de Catalunya through the CERCA Program (https://cerca.cat/). All authors carried out the research independently of the funding body. The findings and conclusions in this manuscript reflect the opinions of the authors alone. The funders had no role in study design, data collection and analysis, decision to publish, or preparation of the manuscript.

**Competing interests:** The authors have declared that no competing interests exist.

**Abbreviations:** DBP, diastolic blood pressure; HDI, Human Development Index; LIC, low-income country; LMIC, low-income and middle-income country; NCD, noncommunicable disease; PURE, Prospective Urban and Rural Epidemiological; REDcap, Research Electronic Data Capture; SBP, systolic blood pressure; UMIC, upper-middle-income country.

hypertension was 30.5% (95% CI, 28.9, 32.0) in urban areas and 27.9% (95% CI, 26.3, 29.6) in rural areas, resulting in a pooled urban-rural difference of 2.45% (95% CI, 1.57, 3.33, I-square: 99.71%, tau-square: 0.00524, $P_{heterogeneity}$ < 0.001). Hypertension prevalence increased over time and the rate of change was greater in rural compared to urban areas, resulting in a pooled urban-rural difference of 5.75% (95% CI, 4.02, 7.48) in the period 1990 to 2004 and 1.38% (95% CI, 0.40, 2.37) in the period 2005 to 2020, $p$ < 0.001 for time period. We observed substantial heterogeneity in the urban-rural difference of hypertension, which was partially explained by urban-rural definition, probably high risk of bias in sampling, country income status, region, and socioeconomic indicators. The urban-rural difference was 5.67% (95% CI, 4.22, 7.13) in low, 2.74% (95% CI, 1.41, 4.07) in lower-middle and −1.22% (95% CI, −2.73, 0.28) in upper-middle-income countries in the period 1990 to 2020, $p$ < 0.001 for country income. The urban-rural difference was highest for South Asia (7.50%, 95% CI, 5.73, 9.26), followed by sub-Saharan Africa (4.24%, 95% CI, 2.62, 5.86) and reversed for Europe and Central Asia (−6.04%, 95% CI, −9.06, −3.01), in the period 1990 to 2020, $p$ < 0.001 for region. Finally, the urban-rural difference in hypertension prevalence decreased nonlinearly with improvements in Human Development Index and infant mortality rate. Limitations included lack of data available from all LMICs and variability in urban and rural definitions in the literature.

## Conclusions

The prevalence of hypertension in LMICs increased between 1990 and 2020 in both urban and rural areas, but with a stronger trend in rural areas. The urban minus rural hypertension difference decreased with time, and with country-level socioeconomic development. Focused action, particularly in rural areas, is needed to tackle the burden of hypertension in LMICs.

---

## Author summary

### Why was this study done?

- Hypertension is one of the main risk factors for morbidity and mortality worldwide.

- Urbanisation is a dynamic process that is occurring mainly in low-income and middle-income countries (LMICs) nowadays. Whether urban-rural differences in hypertension prevalence vary by region, country-level income status, calendar time, or socioeconomic indicators is largely unknown yet important for understanding the public health implications of urbanisation.

### What did the researchers do and find?

- We performed a systematic database search, and after standardised study selection, data extraction, and risk of bias assessment, we analysed 299 surveys including information from over 19.7 million individuals in 66 LMICs.

- We observed a slightly higher prevalence of hypertension in urban compared with rural areas in a meta-analysis. The urban-rural difference varied with urbanisation stage and socioeconomic development, and decreased over time as prevalence in rural areas converged with, and eventually overtook, that of urban areas.

### What do these findings mean?

- The prevalence of hypertension in LMICs has increased over the past 2 decades; the rate of change appears greater in rural compared to urban areas. Overall patterns in the urban-rural difference indicate that as country-level socioeconomic indicators improved, hypertension in rural began to surpass that of urban areas.

- These results have important implications for public health planning: Tackling the global burden of hypertension will require targeted action, particularly in rural areas of LMICs, where there are important opportunities for prevention in the face of socioeconomic development and urbanisation.

## Introduction

Hypertension is a key risk factor for death and disability worldwide [1,2]. In 2019, high systolic blood pressure was the leading risk factor for mortality globally, accounting for 19.2% of all deaths in 2019 [1]. The increase in the burden of noncommunicable diseases (NCDs), including hypertension [3,4], has been larger in low-income and middle-income countries (LMICs) compared to high-income countries in the past 3 decades [3–8], and ischaemic heart disease and stroke were ranked third and fourth causes of death in low-income countries (LICs) and first and second in lower- and upper-middle-income countries in 2019 [9].

The majority of the global population now lives in urban areas (55.7% in 2019 according to the World Bank); however, the transition from rural to urban areas—urbanisation—is occurring mostly in LMICs, specifically in Africa and Asia [10]. Compared to rural areas, urban areas generally provide better access to healthcare, improved water supply and sanitation, and clean household energy, among other attributes that promote health [11]. However, urban areas often concentrate health risks including increased ambient air pollution, low levels of physical activity, and lack of access to high-quality, affordable food [11,12]. Urban areas have therefore been a particular focus of research on prevention of NCDs. However, urbanisation-driven changes associated with high blood pressure and hypertension [5,12] (e.g., shifts from physically demanding to sedentary occupations and increased access to processed foods) [13–15] often occur more rapidly in rural compared to urban areas [16]. A pooled analysis of trends in urban-rural differences in body mass index between 1985 and 2017 showed that the fastest increase in obesity nowadays comes from rural, rather than urban areas [13]. In high-income countries, the higher prevalence of hypertension and other cardiovascular risk factors in rural areas compared to urban areas has been shown, such as in the US [17–19] and in the Prospective Urban and Rural Epidemiological (PURE) multicountry study [20].

Previous global systematic reviews on hypertension have not focused on urban-rural differences in prevalence [3,4,21]. Others included studies without clear urban-rural contrasts, compared rural and urban populations from different studies using different sampling schemes, or

were focused on specific countries and regions [3,21–25]. Consequently, urban-rural differences in hypertension prevalence according to time, country-level stage of urbanisation, and socioeconomic development remain inadequately characterised [7,26,26]. Detailed characterisation of these relationships is needed to support interventions to mitigate the harmful effects of raised blood pressure, a modifiable risk factor for cardiovascular mortality. We hypothesised that urban-rural differences in prevalence of hypertension in LMICs decreased with increasing country-level socioeconomic development and stage of urbanisation [20–22]. We systematically reviewed studies from LMICs between 1990 and 2020 that simultaneously evaluated the prevalence of hypertension in urban and rural areas.

## Methods

This study is reported as per the Preferred Reporting Items for Systematic Reviews and Meta-Analyses (PRISMA) guideline (S1 PRISMA Checklist).

### Search strategy and selection criteria

Detailed methods are available in the S1 Protocol. In brief, a systematic search was carried out in PubMed, Web of Science, Scopus, and Embase in May 2018. We updated the search on March 2022. We set the time limit from 01/01/1990 to 01/05/2018 in the first search round, and from 01/05/2018 to 10/03/2022 in the second search round. We set language to English. We used a range of search terms relating to hypertension, urbanisation, and LMICs. Hand searching was done using citations and reference lists of the included studies and previously published systematic reviews.

We included population-based observational studies. Eligibility criteria for inclusion were as follows:

(a) participants 15 years and older; (b) general population representative of the target population, selected by using a valid sampling technique (e.g., random sampling, multistage sampling, self-weighted sampling, WHO Steps); (c) 400 or more participants [22]; (d) data from LMICs as classified by the World Bank in 2018 fiscal year; (e) data collected from 1990 onwards (period of data collection); (f) prevalence in urban and rural areas evaluated using similar sampling protocols and blood pressure measurements with no more than 4 years of difference between urban and rural measurements; and (g) hypertension definition included measurement of systolic blood pressure (SBP) as $\geq$140 mm Hg and/or diastolic blood pressure (DBP) as $\geq$90 mm Hg and could include use of antihypertensive medication, self-reported diagnosis, or both. When the study did not report hypertension as $\geq$140/90 mm Hg (e.g., old WHO criteria as 160/95 mm Hg or only SBP/DBP means) and we obtained enough information for conversion, we applied validated equations to derive the prevalence based on $\geq$140/90 mm Hg [4].

OTR and AK conducted the literature search. OTR and AK screened all titles, abstracts, and full manuscripts that met inclusion criteria; screening was blinded and implemented using the web platform COVIDENCE. Disagreements were resolved by consensus between OTR, AK and the senior author (CT). Our study protocol was registered in the PROSPERO database (CRD42018091671).

### Data extraction and risk of bias assessment

We developed a standardised electronic data collection form in REDcap (Research Electronic Data Capture), which was piloted by OR, AK, and CT. Data extraction followed a prespecified protocol and was conducted independently by each extractor blinded to extraction data of other extractors. All data extractors (OTR, AK, CDG, FAO, JIH, AC, CT) were trained by the

first author (OTR) and paired in 4 teams of 2 data extractors each (OTR/AK, CDG/JIH, OTR/ FAO, AC/CT). Countries were randomly assigned to each pair. The first 10 papers were piloted by each data extractor for clarifications and refinement. If a study reported more than 1 survey (e.g., different years or different countries), we extracted data for all surveys. If there was more than 1 paper from the same cohort/survey, we included the paper providing the most comprehensive and clear information on hypertension prevalence for urban-rural contrast. When information was not available in the main paper, we used data from additional papers from the same cohort and supplementary data cited in the main paper (e.g., raw data publicly available, WHO STEPS Country Reports) for extracting relevant information. We extracted crude and adjusted prevalence estimates when both estimates were available. We extracted standard errors for the prevalence of hypertension following the hierarchy (1) standard error when provided; (2) lower and upper limit values from confidence intervals [27]; and (3) square root of ([hypertension prevalence × (1—hypertension prevalence)] / sample size) [3]. We also extracted blood pressure and sex-specific data when available. Data were then exported for standardisation and resolving conflicts. Disagreements between pairs were checked by OTR and AK and conflicts discussed with CT.

We used the OHAT Bias Tool to evaluate the risk of bias [28]. We evaluated 3 domains: "Selection Bias—Sampling", "Detection bias/Measurement error—Exposure", and "Detection bias/Measurement error—Outcome". We chose these domains because of their relevance to our research objective: to assess prevalence (sampling) of hypertension (outcome) in 2 contrasting areas (exposure). The details of the OHAT Bias Tool are available in the S1 Protocol.

### Country-level socioeconomic data

We predefined 5 country-level socioeconomic indicators, calendar year, and country region to be evaluated in the meta-regression. We adjusted for these country-level socioeconomic indicators in the meta-regression because our hypothesis was that socioeconomic development is correlated with calendar time and the main drivers of hypertension prevalence in an area, such as urbanisation, diet, and physical activity. We extracted the region, historical income classification, infant mortality rate, GNI per capita, Atlas method (current US$), and proportion of urban population from the World Bank, and the Human Development Index (HDI) from the United Nations Development Programme (S1 Protocol). We selected indicators that were available yearly for the entire period and with global coverage, capturing socioeconomic development (i.e., HDI, income, GNI), extent of urbanisation (i.e., proportion of urban population), and proxies of population healthcare (i.e., infant mortality rate) [29]. We extracted yearly country-level indicators from 1990 to 2020 and matched them to the corresponding year of the start of data collection for each survey.

### Data analysis

We conducted the meta-analysis and meta-regression for the entire period (1990 to 2020) and for 2 periods (1990 to 2004 and 2005 to 2020), using the time-period as a moderator. In all instances, year was defined as the year when the data collection started. The cutpoint between the 2 periods was based on the median year between 1990 and 2019. We prioritised age, sex, and/or sampling weight adjusted prevalence estimates instead of crude estimates when both were available. The unit of analysis was the survey. Our primary outcome was urban-rural prevalence difference.

We conducted the meta-analysis for hypertension prevalence, urban-rural prevalence difference, urban-rural average blood pressure, and meta-regression for the urban-rural prevalence difference using a random-effects model, with a restricted-maximum likelihood

estimator and applying the Knapp and Hartung adjustment.[30] As a post hoc sensitivity analysis for the urban-rural prevalence difference, we fit a meta-analytic multivariate random-effects model, accounting for each survey as a random intercept as in the main analysis, but adding other random intercept accounting for the country.[31] As a sensitivity analysis for the meta-analysis of hypertension prevalence, we fit a model with a generalised mixed model [32,33]. We estimated pooled urban-rural differences in hypertension prevalence across the 2 time periods, country income classification (3 categories), and country region (6 categories) using a meta-regression model [34]. We estimated $I^2$ and $tau^2$ to evaluate heterogeneity. In the meta-regression, we used $R^2$ to estimate the amount of heterogeneity accounted for by the moderators [27,35–37]. Publication bias was evaluated with the Egger test. We evaluated non-linearity for continuous moderators applying restricted cubic splines and choosing the most parsimonious model based on AIC, BIC, and likelihood-ratio test.

We adjusted all meta-regression models with the 5 study-level moderators that explained part of the heterogeneity when evaluating country-level moderators: use of groups to define urban and rural areas, number of blood pressure readings, sampling bias, detection bias (urban/rural), and whether the prevalence was adjusted by age-sex/sampling weights. In the meta-regression models, we entered socioeconomic country-level moderators separately in each model because of high collinearity (S1 Protocol). We derived the predicted urban-rural difference from each meta-regression model, setting each moderator to vary within the observed range (e.g., HDI from 0.30 to 0.85) and setting the 5 study-level characteristics to its expected least-biased category (e.g., probably low risk of bias in sampling). In a post hoc decision to understand the main driver of variation in the urban-rural prevalence differences according to country-level indicators (i.e., the change in difference was due to change in prevalence in urban, rural, or both areas), we derived the predicted hypertension prevalence from each meta-regression model for urban and rural areas following the same steps for the urban-rural difference.

All analyses were done in R, version 4.0.2, using the packages *tidyverse*, *rms*, and *metafor* [36,38,39]. Any deviance from the prespecified analysis was labelled as post hoc. All statistical tests were two-sided and a $P \leq 0.05$ was considered statistically significant.

## Results

### Study selection

From the 18,951 retrieved records, 1,309 full-text articles were assessed for eligibility after title/abstract screening (Fig 1). We included 255 studies reporting 299 surveys, covering 66 LMICs and 6 regions (S1 Data). The total sample size was 19,770,946 participants (mean age 45.4 ± SD = 9 years, 53.0% females and 63.1% from rural areas). Countries with the most surveys were China (*n* = 66, 22%), India (*n* = 39, 13%), Iran (*n* = 17, 6%), Bangladesh (*n* = 13, 4%), Nigeria (*n* = 11, 4%), and Vietnam (*n* = 10, 3%) (S1 Data). The coverage of data collection over time is shown on S1 Data. General characteristics of the 299 surveys are shown in Table 1. Additional information on studies' and surveys' characteristics is in S2 Data.

### Hypertension prevalence

The overall prevalence of hypertension in urban areas was 30.5% (95% CI, 28.9 to 32.0, I-square: 99.95%, tau-square: 0.01815, $P_{heterogeneity} < 0.001$) and 27.9% (95% CI, 26.3 to 29.6, I-square: 99.97%, tau-square: 0.01984, $P_{heterogeneity} < 0.001$) in rural areas, with similar pattern by sex (S3 Data). The overall rural prevalence of hypertension across country income ranged from 21.9% (95% CI, 19.3 to 24.6) in LICs to 36.3% (95% CI, 33.6 to 39.1) in upper-middle-income countries (UMICs). The overall urban prevalence of hypertension ranged from 27.7%

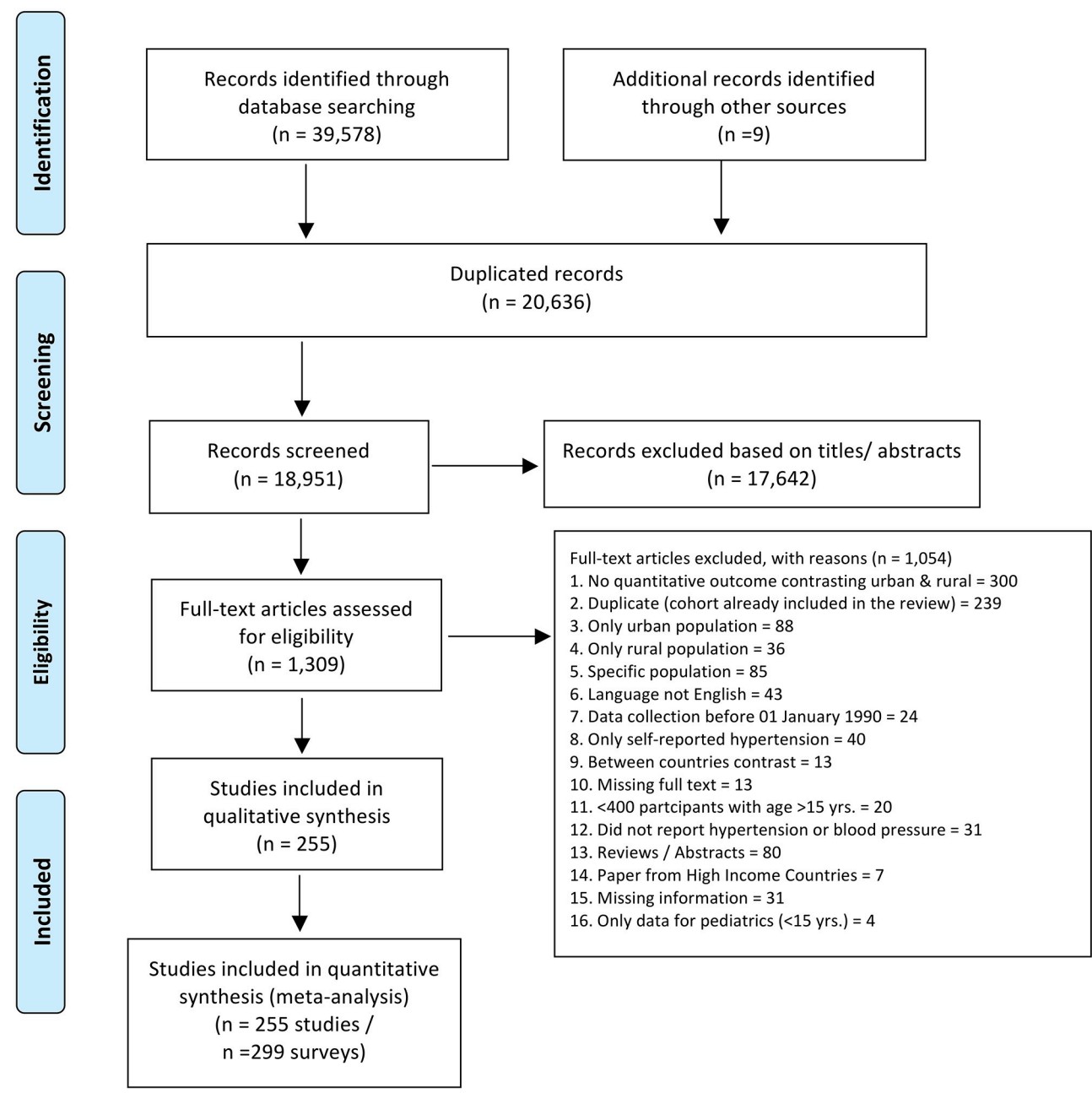

**Fig 1. PRISMA flowchart.** Study flowchart after systematic search between 01/01/1990 and 10/03/2022 in the PubMed, Web of Science, Scopus, and Embase databases.

(95% CI, 25.0 to 30.4) in LIC to 35.1% (95% CI, 32.3 to 37.9) in UMIC (S3 Data). The overall rural prevalence of hypertension across country region ranged from 23.8% (95% CI, 20.3 to 27.3) in South Asia to 43.8% (95% CI, 37.8% to 49.8%) in Europe and Central Asia. The overall urban prevalence of hypertension across country region ranged from 27.6% (95% CI, 22.3 to 32.9) in Middle East and Central Asia to 37.7% (95% CI, 31.7% to 43.7%) in Europe and Central Asia (S3 Data).

The prevalence of hypertension increased for rural areas, while it increased, decreased, or remained stable for urban areas when stratified by sex, income, and region (S3 Data). When

**Table 1. Overall characteristics of the 299 surveys included from 66 LMICs.**

| Characteristic | Description |
|---|---|
| **Region** | |
| East Asia and Pacific | 96 (32%) |
| Sub-Saharan Africa | 73 (24%) |
| South Asia | 59 (20%) |
| Middle East and North Africa | 26 (9%) |
| Europe and Central Asia | 20 (7%) |
| Latin America and Caribbean | 25 (8%) |
| **Income classification at start of data collection** | |
| Low-income | 96 (32%) |
| Lower-middle-income | 114 (38%) |
| Upper-middle-income | 89 (30%) |
| **Coverage** | |
| National | 104 (35%) |
| Subnational | 42 (14%) |
| Other | 153 (51%) |
| **Rural and Urban definition**[*] | |
| National definition (e.g., census) | 199 (67%) |
| Quantitative metric (e.g., distance, density) | 76 (25%) |
| Specific score | 4 (1%) |
| Specific population groups (e.g., farmers, indigenous) | 24 (8%) |
| **Hypertension and Blood pressure measurements**[*] | |
| Hypertension prevalence included self-reported | 80 (27%) |
| Hypertension prevalence included taking antihypertensive drug | 223 (75%) |
| Number of blood pressure readings per visit $\geq$2 ($n$ = 24 not reported) | 273/275 (99%) |
| Manual blood pressure device ($n$ = 30 not reported) | 96/269 (36%) |
| **Risk of bias (probably high risk)**[*] | |
| Sampling | 47 (16%) |
| Exposure | 56 (19%) |
| Outcome | 14 (5%) |
| Any domain | 89 (30%) |
| **Starting year of data collection, median [p25-p75]; (min-max)** | 2009 [2005–2013]; (1990–2019) |
| **Ending year of data collection, median [p25-p75]; (min-max)** | 2010 [2006–2014]; (1990–2020) |
| **Total sample size, median [p25-p75]; (min-max)** | 4,376 [2,018–10,280]; (416–9,745,640) |

[*]Each study can have more than 1 category, i.e., the total sum could add more than 100%.

LIC, low-income country; LMIC, lower-middle-income country; UMIC, upper-middle-income country.

comparing the period from 1990 to 2004 ($n$ = 70) to 2005 to 2020 ($n$ = 221), the prevalence in rural areas increased from 23.8% (95% CI, 20.5 to 27.1) to 29.3% (95% CI, 27.4 to 31.1) ($p$ = 0.005 for time-period, R-square: 2.47%), while in urban areas remained stable, from 29.8% (95% CI, 26.6 to 33.0) to 30.7% (95% CI, 28.9 to 32.5) ($p$ = 0.612 for time-period, R-square: 0.00%). The sensitivity analysis for the meta-analysis of hypertension prevalence yielded similar results (S3 Data).

## Urban-rural prevalence difference

The pooled difference between urban and rural areas was 2.45% (95% CI, 1.57 to 3.33, I-square: 99.71%, tau-square: 0.00524, $P_{heterogeneity} < 0.001$) (Table 2). This difference was

**Table 2. Difference in hypertension prevalence between urban and rural areas from the 299 surveys included from 66 LMICs.**

| Period | Category | n | Urban-rural prevalence difference (95% CI) | $I^2$ | $tau^2$ | *P* value for heterogeneity | $R^2$ | *P* value for moderator |
|---|---|---|---|---|---|---|---|---|
| **Overall** | | | | | | | | |
| Period 1990–2020* | All | 291 | 2.45% (1.57, 3.33) | 99.71% | 0.00524 | <0.001 | | |
| **Overall by period** | | | | | | | | |
| Period 1990–2004* | All | 70 | 5.75% (4.02, 7.48) | 99.63% | 0.00490 | <0.001 | 6.40% | <0.001 |
| Period 2005–2020* | All | 221 | 1.38% (0.40, 2.37) | | | | | |
| **Income status at data collection** | | | | | | | | |
| Period 1990–2020 | LIC | 96 | 5.67% (4.22, 7.13) | 99.46% | 0.00465 | <0.001 | 13.67% | <0.001 |
| Period 1990–2020 | LMIC | 114 | 2.74% (1.41, 4.07) | | | | | |
| Period 1990–2020 | UMIC | 89 | −1.22% (−2.73, 0.28) | | | | | |
| **Region** | | | | | | | | |
| Period 1990–2020 | East Asia and Pacific | 96 | 0.50% (−0.87, 1.86) | 99.53% | 0.00420 | <0.001 | 21.97% | <0.001 |
| Period 1990–2020 | sub-Saharan Africa | 73 | 4.24% (2.62, 5.86) | | | | | |
| Period 1990–2020 | South Asia | 59 | 7.50% (5.73, 9.26) | | | | | |
| Period 1990–2020 | Middle East and North Africa | 26 | 0.72% (−1.93, 3.36) | | | | | |
| Period 1990–2020 | Europe and Central Asia | 20 | −6.04% (−9.06, −3.01) | | | | | |
| Period 1990–2020 | Latin America and Caribbean | 25 | 2.20% (−0.57, 4.97) | | | | | |

*For the overall estimates, we used the original PURE study containing data from 14 LMICs countries (*n* = 126,624 participants, 14 surveys). For stratified analysis by income and region, we used data from 7 studies that reported data at country level for 9 countries from the original PURE study (*n* = 104,196 participants, 9 surveys).

LIC, low-income country; LMIC, lower-middle-income country; UMIC, upper-middle-income country.

greater for the period 1990 to 2004 (5.75%, 95% CI, 4.02 to 7.48), compared with the period 2005 to 2020 (1.38%, 95% CI, 0.40 to 2.37) (*p* < 0.001 for time period, R-square: 6.40%). The pooled difference between urban and rural areas varied by country income status: 5.67% (95% CI, 4.22 to 7.13) in LICs, 2.74% (95% CI, 1.41 to 4.07) in LMICs, and −1.22% (95% CI, −2.73 to 0.28) in UMICs (*p* < 0.001 for overall income effect, R-square: 13.67%). For the period 2005 to 2020 compared to 1990 to 2004, the urban-rural difference decreased for LICs (8.16% versus 3.87%), remained stable for LMICs (2.27% versus 2.90%), and decreased for UMICs (9.26% versus −1.72%) (S4 Data). The pooled difference between urban and rural areas varied across regions (*p* < 0.001 for overall region effect, R-square: 21.97%) (Table 2). The urban-rural difference was highest for South Asia (7.50%, 95% CI, 5.73 to 9.26), followed by sub-Saharan Africa (4.24%, 95% CI, 2.62 to 5.86). Studies from Europe and Central Asia region showed higher prevalence in rural than urban areas (−6.04%, 95% CI, −9.06 to −3.01).

We obtained average blood pressure from 105 surveys (35%, 105/299). The distributions of SBP and DBP stratified by urban and rural areas are in S5 Data. Mean blood pressure followed the same patterns as urban-rural prevalence difference overall and by income and region. Pooled SBP was 126.2 mm Hg (95% CI, 124.7 to 127.7) in urban areas compared with 125.2 mm Hg (95% CI, 123.6 to 126.8) in rural areas (*n* = 105, mean difference 0.99 mm Hg, 95% CI, −0.03 to 2.02). Pooled DBP was 79.1 mm Hg (95% CI, 78.2 to 80.0) in urban areas compared with 77.9 mm Hg (95% CI, 77.0 to 78.8) in rural areas (*n* = 105, mean difference 1.11 mm Hg, 95% CI, 0.51 to 1.70). The urban-rural difference was greater for the first period (1990 to 2004) for SBP: 2.34 mm Hg (95% CI, 0.60 to 4.09) compared to 0.30 mm Hg (95% CI, −0.95 to 1.55) in the second period (2005 to 2020) (S5 Data).

When considering the 299 surveys, the pooled difference between urban and rural areas was 2.49% (95% CI, 1.61 to 3.37, I-square: 99.71%, tau-square: 0.00540, $P_{heterogeneity} < 0.001$). There was no evidence for publication bias (Egger test, $p = 0.06$). In the sensitivity analysis using a multivariate random-effects model accounting for study and country, we observed broadly comparable estimates with the main analysis (Overall: 1.82%, 95% CI, 0.47 to 3.16) and when accounting for the time period, income status, and region (S6 Data).

In univariate analyses evaluating study characteristics as moderators, use of population subgroups to define urban and rural areas (R-square: 1.33%), number of blood pressure readings (R-square: 1.24%), prevalence adjusted by age/sex/sampling weights (R-squared: 0.90%), probably high risk of bias in sampling (R-square: 0.95%), and probably high risk of bias in urban-rural definition (R-square: 0.91%) explained part of the heterogeneity (S6 Data). For instance, 47 surveys were classified as probably high risk of bias in sampling, yielding an urban-rural difference in the prevalence of 4.48% (95% CI, 2.22 to 6.73) compared with 2.14% (95% CI, 1.19 to 3.09) from 252 studies with low risk of bias in sampling ($p = 0.061$ for bias in sampling as moderator). Fifty-six surveys were classified as probably high risk of bias in the domain of urban-rural definition, yielding to an urban-rural difference in the prevalence of 0.70% (95% CI, −1.35 to 2.75) compared with 2.89% (95% CI, 1.92 to 3.86) from 243 surveys with low risk of bias in sampling ($p = 0.058$ for bias in urban-rural definition as moderator). There was no evidence for moderation ($p = 0.715$) between surveys that used self-reported hypertension diagnosis ($n = 80$, 2.22%, 95% CI, 0.53 to 3.92) and surveys did not account for self-reported hypertension diagnosis ($n = 219$, 2.59%, 95% CI, 1.56 to 3.62) (S6 Data).

The urban-rural difference in prevalence of hypertension adjusted for use of population subgroups to define urban and rural areas, number of blood pressure readings, prevalence adjusted by age/sex/sampling weights, probably high risk of sampling bias, and probably high risk of bias in urban-rural definition was 2.50% (95% CI, 1.65 to 3.34).

From the meta-regression, the model with the 5 study-level moderators (model-2, S6 Data) accounted for 7.2% of the heterogeneity. After including separately each country-level characteristic to the model-2, country region (model-3, R-square: 26.1%, S6 Data), infant mortality rate (model-12, R-square: 26.1%, S6 Data), HDI (model-10, R-square: 23.2%, S6 Data), proportion of urban population (model 11, R-square: 21.2%, S6 Data), GNI per capita (model-9, R-square: 20.8%, S6 Data), and country income classification (model-4, R-square: 19.1%, S6 Data) explained further proportion of the heterogeneity. The model with the year upon starting data collection explained 9.8% of the heterogeneity (model-5, S6 Data).

Overall, the urban-rural difference in prevalence of hypertension decreased with increasing calendar time (model-2), HDI (model-10), proportion of urban population (model-11), and GNI per capita (model-9) while it increased with increasing infant mortality rate (model-12) (Figs 2 and 3 and 4). Downward trends over time in the urban-rural prevalence difference varied between regions (S6 Data); downward trends were steepest in Latin America and Caribbean, East Asia and Pacific, and Middle East and North Africa compared with South Asia, sub-Saharan Africa, and Europe and Central Asia. The difference in urban-rural hypertension prevalence varied nonlinearly with HDI, infant mortality rate, and proportion of urban population (Figs 3 and 4). Hypertension prevalence in rural areas varied according to these indicators more than in urban areas. For instance, there was a steeper increase by year (Fig 2), HDI (Fig 3), and GNI per capita (Fig 4) in rural areas compared with urban areas.

## Discussion

Our systematic review and meta-analysis including data from 299 surveys across 66 LMICs for the 1990 to 2020 period resulted in several key findings. We observed an overall prevalence of

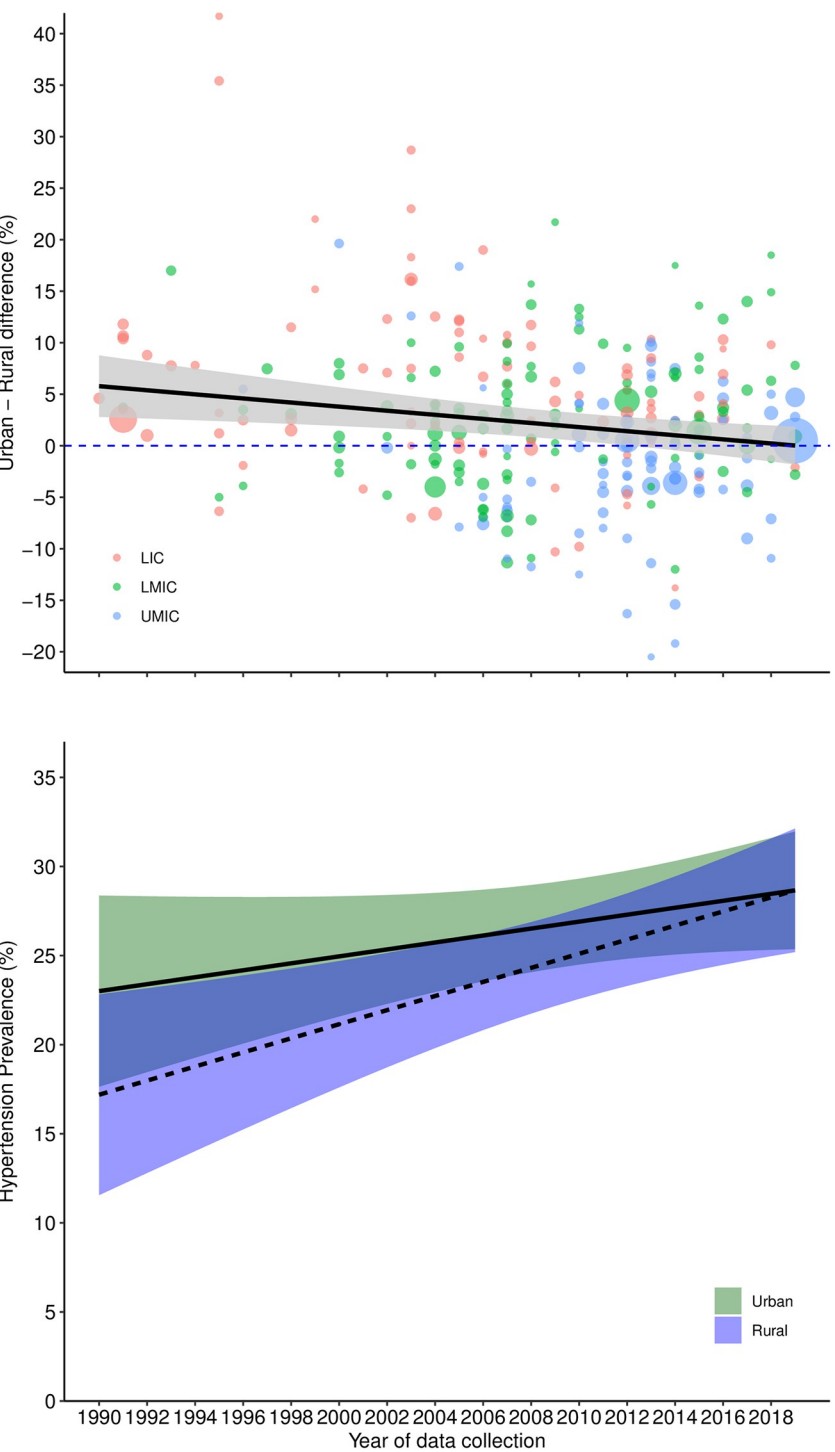

**Fig 2. Urban-rural difference in hypertension prevalence from 1990 to 2020.** Predicted urban-rural differences from a meta-regression model with year of start of data collection and 5 study-level features: use of population groups to define urban and rural areas, number of blood pressure readings, sampling bias, detection bias (urban/rural), and prevalence adjusted by age/sex/sampling weights. The plot shows the prediction (marginal mean from model-5) of this model varying year from 1990 to 2019 (year starting data collection) and setting the 5 study-level features to the least biased category (no use of groups, ≥2 readings, probably low risk of sampling bias, probably low risk of detection bias (urban/rural), and adjusted prevalence). Income country status is only to illustrate each survey, and it is not adjusted in the model. Shaded areas represent 95% confidence interval and circle sizes proportional to inverse of variance. In the bottom panel, the solid lines represent urban areas and dashed lines represent rural areas. LIC, low-income country; LMIC, lower-middle-income country; UMIC, upper-middle-income country.

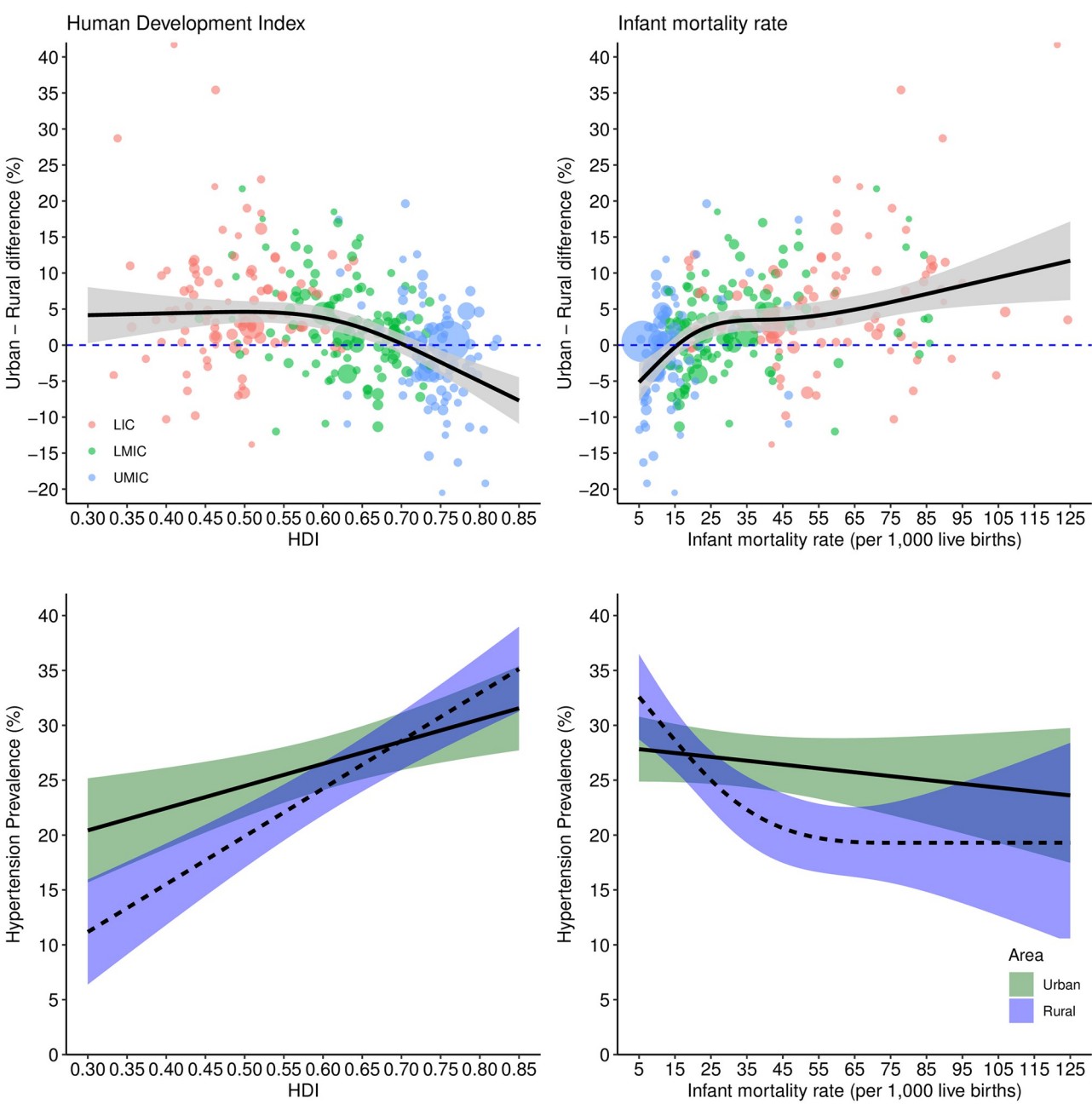

**Fig 3. Urban-rural difference in hypertension prevalence and prevalence of hypertension in urban and rural areas according to HDI and infant mortality rate.** Predicted urban-rural differences and prevalence from meta-regression models with HDI or infant mortality rate, and 5 study-level features: use of population groups to define urban and rural areas, number of blood pressure readings, sampling bias, detection bias (urban/rural), and prevalence adjusted by age/sex/sampling weights. The plot shows the prediction (marginal mean from models 10 and 12) of this model varying HDI from 0.3 to 0.85 and infant mortality rate from 5 to 125 (range from observed data) and setting the 5 study-level features to the least biased category (no use of groups, ≥2 readings, probably low risk of sampling bias, probably low risk of detection bias (urban/rural), and adjusted prevalence). Income country status is only to illustrate each survey, and it is not adjusted in the model. Shaded areas represent 95% confidence interval and circle sizes proportional to inverse of variance. In the bottom panel, the solid lines represent urban areas and dashed lines represent rural areas. HDI, Human Development Index (higher values denote more development); LIC, low-income country; LMIC, lower-middle-income country; UMIC, upper-middle-income country.

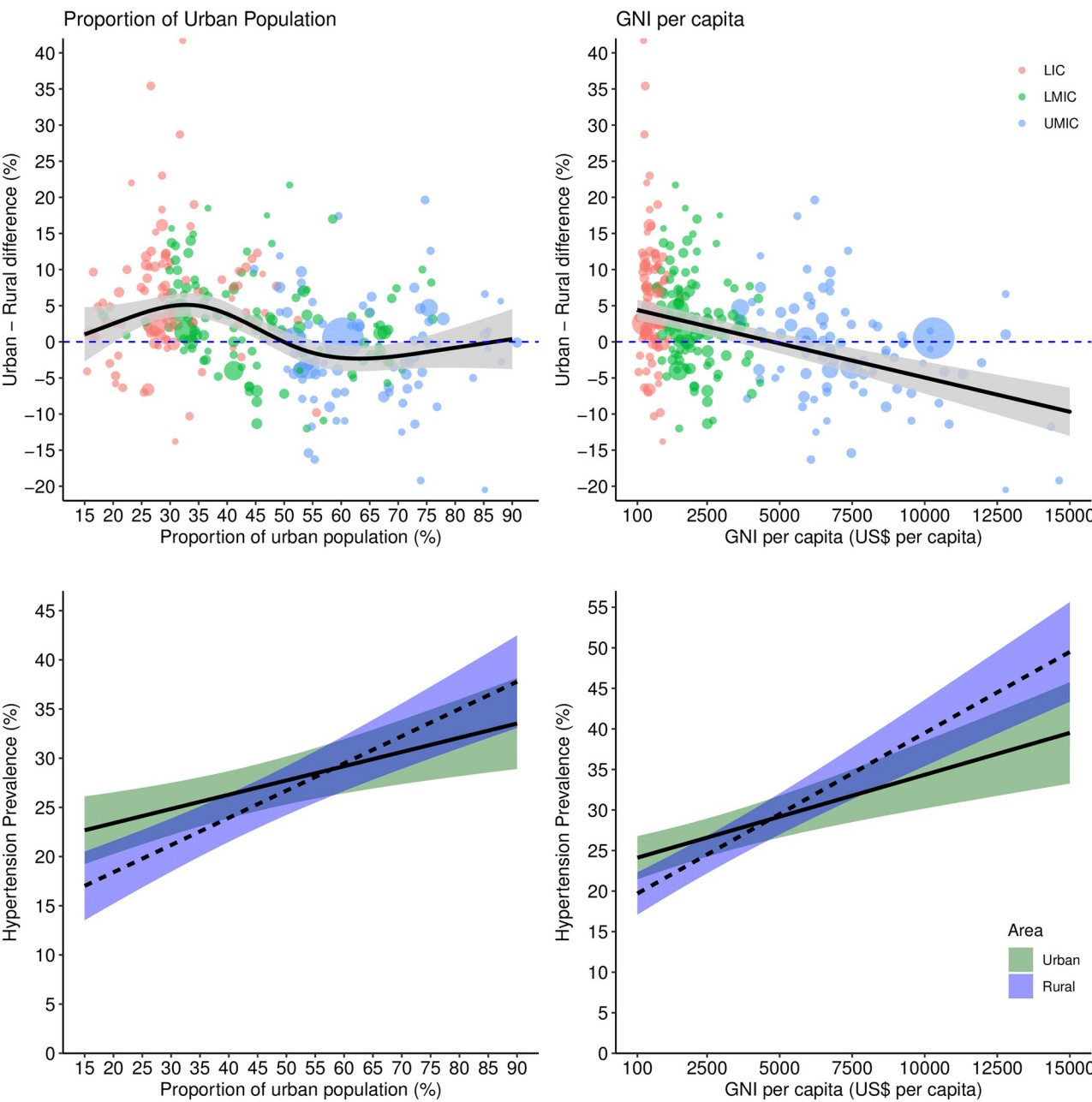

**Fig 4. Urban-rural difference in hypertension prevalence and prevalence of hypertension in urban and rural areas according to proportion of urban population and GNI per capita.** Predicted urban-rural differences and prevalence from meta-regression models with proportion of urban population or GNI per capita, and 5 study-level features: use of population groups to define urban and rural areas, number of blood pressure readings, sampling bias, detection bias (urban/rural), and prevalence adjusted by age/sex/sampling weights. The plot shows the prediction (marginal mean from models 9 and 11) of this model varying proportion of urban population from 15% to 90% and GNI per capita from $100 to $15,000 (range from observed data) and setting the 5 study-level features to the least biased category (no use of groups, ≥2 readings, probably low risk of sampling bias, probably low risk of detection bias (urban/rural), and adjusted prevalence). Income country status is only to illustrate each survey, and it is not adjusted in the model. Shaded areas represent 95% confidence interval and circle sizes proportional to inverse of variance. In the bottom panel, the solid lines represent urban areas and dashed lines represent rural areas. GNI, gross national income; LIC, low-income country; LMIC, lower-middle-income country; UMIC, upper-middle-income country.

hypertension of 30.5% (95% CI, 28.9 to 32.0) in urban areas and 27.9% (95% CI, 26.3 to 29.6) in rural areas, resulting in a pooled urban-rural difference of 2.45% (95% CI, 1.57 to 3.33). This difference varied according to country-level socioeconomic development supporting our hypothesis. The urban-rural difference in prevalence decreased with increasing development until a point of convergence, after which rural areas had higher hypertension prevalence in the most developed LMICs. This pattern was primarily driven by a more rapid increase in hypertension prevalence in rural compared to urban areas, such that rural areas caught up with and eventually overtook urban areas with increasing level of development.

To the best of our knowledge, this is the first systematic review focused on urban-rural differences in the prevalence of hypertension in LMICs worldwide, estimating the role of country-level indicators in this difference. Previous systematic reviews focused on overall hypertension prevalence, deriving urban and rural prevalence estimates as subgroups, or restricted to specific countries and regions, or lacked clear urban-rural comparisons [3,21–24,40,41]. A previous systematic review evaluating the prevalence of hypertension in LMICs until 2015 including 242 studies and 1,494,609 adults from 45 countries observed a pooled hypertension prevalence of 32.7% (95% CI, 30.4 to 35.0, $n = 80$ studies) in urban areas compared with 25.2% (95% CI, 20.9 to 29.8, $n = 50$ studies) in rural areas, yielding a global difference of 7.5% [21]. However, Sarki and colleagues included studies that reported only urban, only rural, or mixed populations without clear urban and rural contrast, which likely explains their larger pooled estimate compared to ours [21]. Studies including populations from urban areas alone are frequent in the literature (we excluded 88 studies reporting only urban populations compared with 36 reporting only rural). Another limitation of previous literature is that awareness of hypertension diagnosis is higher in urban areas [16,20], which can overestimate hypertension prevalence in urban areas if self-reported diagnosis alone is used to define the outcome. In contrast, we included only studies that simultaneously reported both urban and rural hypertension prevalence using comparable sampling protocols and outcome measurements and excluded studies with self-reported diagnosis without blood pressure measurements. Our review therefore provides more accurate estimates of urban-rural differences in hypertension prevalence by avoiding several potential biases that would overestimate the prevalence in urban areas.

We observed variation in urban-rural differences in hypertension prevalence across the 6 regions. The largest urban-rural difference was observed in South Asia, followed by sub-Saharan Africa, regions that also had the smallest downward trend in the urban-rural difference (S6 Data). These regions also showed an overall increase in mean blood pressure over time, in contrast with regions with decreasing (e.g., Latin America and Caribbean and high-income countries), or stable trends (e.g., Middle East and North Africa), as reported by global estimates of hypertension prevalence [4].

Overall, it seems there has been occurring a convergence of hypertension prevalence between urban and rural areas in the last decade, largely driven by steeper increase in rural compared to urban areas. This phenomenon has been reported in systematic reviews from India and China [25,40,42]. Several factors may explain the steeper trends in hypertension prevalence in rural areas with time, country level of urbanisation, and socioeconomic development [8]. Out-migration by young people could lead to an increasingly older age structure in rural areas. We explored this using data available from 135 surveys from which we could extract mean age from urban and rural areas. Mean age was 47.6 ± 10 years in rural and 46.6 ± 11 years in urban areas. In these 135 surveys, the unadjusted urban-rural difference was 3.91% (95% CI, 2.40 to 5.42) compared to an age-adjusted difference of 4.84% (95% CI, 3.31 to 6.37), suggesting that the older age structure in rural areas partially explains the convergence of hypertension prevalence in urban and rural areas. Additionally, rural areas in LMICs may

suffer from a double burden of increasing levels of ambient air pollution [1] with increasing socioeconomic development on top of high levels of household air pollution due to inefficient fuels and technologies for meeting heating and cooking needs [12]. Urbanisation has been linked to shifts toward more sedentary occupations and unhealthy diets, increasing the risk of hypertension and other NCDs [13–15] This relationship is likely to be nonlinear, reaching a saturation or inflection point at a certain level of urbanicity or development [42,43]. Evidence indicates that physical activity levels in urban areas of high-income countries can be higher than in rural areas [44]. Similarly, the increasing rural burden of NCDs has been recently shown for obesity using individual-level longitudinal data [13,15].

There was substantial heterogeneity in the urban-rural differences of hypertension, which was partially explained by study-level and country-level moderators. A challenge to evaluating statistical heterogeneity is that $I^2$ is not an absolute measure of heterogeneity [37,45,46] and can be influenced by the study size, i.e., the precision on the estimates [46]. Therefore, $I^2$ is expected to be high in our systematic review from population-based studies. Nevertheless, using study- and country-level indicators, we could explain about 20% to 25% of the statistical heterogeneity. Although limited by aggregated data in the meta-analysis and meta-regression, other country or regional indicators could help to further understand the observed heterogeneity, as pooled analysis using individual-level population data [4].

Strengths of this systematic review include protocol registration, inclusion of population-based studies covering a wide time period, and standardised and blinded data extraction. However, our review also has limitations. First, although we used a wide literature search strategy, we did not find data from all LMICs. Additionally, we restricted our search to manuscripts in English and excluded 43 (43/1,309, 3.3%) potential full-texts because of language. Although this could have contributed to underrepresentation of some countries, we do not expect that inclusion of non-English papers would have changed our results. Second, all studies were population-based representing the target population of the sampled areas and half of studies were national or subnational. However, we cannot guarantee that the urban and rural differences we extracted are representative of all urban and rural areas across each country. Third, we evaluated time trends in the meta-regression using repeated cross-sectional surveys. Because different populations could have been included in different years, the temporal analysis relies on the assumption that population-based surveys, using clear urban-rural contrasts and the same criteria for hypertension, yielded comparable prevalence estimations over time. Fourth, there is no standard definition for urban and rural areas, and included studies used different criteria and metrics for defining these areas. However, 67% of studies based their definition on national administrative urban/rural definitions, which makes our results more generalisable at the country level and informative for public health planning. Our review highlights the paucity of multicomponent measures of urbanicity, which may outperform dichotomous measures [47], as a risk factor for hypertension. Fifth, although 99% of the included studies had at least 2 blood pressure measurements, most of them measured blood pressure at only 1 visit, which could overestimate the prevalence of hypertension. We addressed this limitation by including the number of blood pressure measurements as a moderator of our primary outcome and adjusting for it in the meta-regression models. Finally, our review protocol did not include high-income countries, which could have provided additional insight into urban-rural difference in hypertension prevalence across the full range of development. We speculate, based on findings from the PURE studies showing higher prevalence of hypertension in rural compared with urban areas in high-income countries (36.4% for urban, 40.2% for rural, difference −3.8%), that we would have seen an even larger negative urban-rural difference for high-income countries than what we observed for upper-middle-income countries (35.1% for urban, 36.3% for rural, difference −1.22%) [20].

Our results have important public health implications. Our results reinforce the need of preventive and control measures for hypertension in LMICs, with a special focus on rural areas [48], where the levels of awareness, treatment, and control of hypertension remain considerably lower compared to urban areas [20,42].

In conclusion, our results challenge the accepted wisdom that urban areas in particular should be a focal point for prevention of hypertension. Using an analysis designed to rigorously compare differences in hypertension prevalence in urban compared to rural areas in LMICs from 1990 to 2020, we observed that overall, this difference is fairly modest and dependent on several factors including, time, global region, and country-level socioeconomic development. Our results indicate stronger trends in hypertension prevalence in rural compared to urban areas with time and socioeconomic development, resulting in convergence of hypertension prevalence after which prevalence in rural areas is higher. More attention is warranted on rural areas as important targets for efforts to decrease the global burden of hypertension through reduction of risk factors, increasing awareness, and hypertension control.

## Supporting information

**S1 PRISMA Checklist. PRISMA checklist.**
(PDF)

**S1 Protocol. Protocol and additional methods.**
(PDF)

**S1 Data. Number and geographical country coverage of included studies.**
(PDF)

**S2 Data. Characteristics, risk of bias, and summary of the 255 studies and 299 surveys.**
(PDF)

**S3 Data. Meta-analysis and meta-regression results for prevalence of hypertension.**
(PDF)

**S4 Data. Urban-rural hypertension prevalence difference by sex, income, and region.**
(PDF)

**S5 Data. Systolic and diastolic blood pressure distribution and urban-rural differences.**
(PDF)

**S6 Data. Additional meta-analysis and meta-regression models results.**
(PDF)

## Author Contributions

**Conceptualization:** Otavio T. Ranzani, Anjani Kalra, Cathryn Tonne.

**Data curation:** Otavio T. Ranzani, Anjani Kalra, Chiara Di Girolamo, Ariadna Curto, Fernanda Valerio, Jaana I. Halonen, Cathryn Tonne.

**Formal analysis:** Otavio T. Ranzani, Xavier Basagaña, Cathryn Tonne.

**Funding acquisition:** Cathryn Tonne.

**Methodology:** Otavio T. Ranzani, Anjani Kalra, Chiara Di Girolamo, Ariadna Curto, Fernanda Valerio, Jaana I. Halonen, Xavier Basagaña, Cathryn Tonne.

**Project administration:** Cathryn Tonne.

**Resources:** Cathryn Tonne.

**Supervision:** Cathryn Tonne.

**Validation:** Xavier Basagaña.

**Writing – original draft:** Otavio T. Ranzani.

**Writing – review & editing:** Anjani Kalra, Chiara Di Girolamo, Ariadna Curto, Fernanda Valerio, Jaana I. Halonen, Xavier Basagaña, Cathryn Tonne.

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
