## [Editor Report · Decision Letter 0]

22 Sep 2021

Dear Dr Ranzani, 

Thank you for submitting your manuscript entitled "Urban-rural differences in hypertension prevalence of low-income and middle-income countries: a systematic review and meta-analysis from 60 countries and 8.5 million participants" for consideration by PLOS Medicine.

Your manuscript has now been evaluated by the PLOS Medicine editorial staff and I am writing to let you know that we would like to send your submission out for external peer review.

Please re-submit your manuscript within two working days, i.e. by Sep 24 2021 11:59PM.

Kind regards,

Beryne Odeny

Associate Editor

PLOS Medicine

---

## [Decision Letter · Decision Letter 1]

9 Mar 2022

Dear Dr. Ranzani,

Thank you very much for submitting your manuscript "Urban-rural differences in hypertension prevalence of low-income and middle-income countries: a systematic review and meta-analysis from 60 countries and 8.5 million participants" (PMEDICINE-D-21-03953R1) for consideration at PLOS Medicine. 

Your paper was evaluated by a senior editor and discussed among all the editors here. It was also sent to independent reviewers, including a statistical reviewer. The reviews are appended at the bottom of this email and any accompanying reviewer attachments can be seen via the link below:

[LINK]

In light of these reviews, I am afraid that we will not be able to accept the manuscript for publication in the journal in its current form, but we would like to consider a revised version that addresses the reviewers' and editors' comments. Obviously we cannot make any decision about publication until we have seen the revised manuscript and your response, and we plan to seek re-review by one or more of the reviewers. 

We expect to receive your revised manuscript by Mar 30 2022 11:59PM. Please email us (plosmedicine@plos.org) if you have any questions or concerns.

We look forward to receiving your revised manuscript. 

Sincerely,

Beryne Odeny, 

PLOS Medicine

plosmedicine.org

1) Please revise your title according to PLOS Medicine's style. Your title must be nondeclarative and not a question. It should begin with main concept if possible. Please place the study design only (for example, “A systematic review and meta-analysis”) in the subtitle (i.e., after a colon).

2) Please include line numbers in your next draft

3) Abstract:

a) Please ensure that all numbers presented in the abstract are present and identical to numbers presented in the main manuscript text.

b) Please provide both 95% CIs and p values where appropriate.

4) The Data Availability Statement (DAS) requires revision. For each data source used in your study:

5) Author Summary – Please reduce the length of bullet points to no more than 4 lines per point.

6) Thank you for providing the PRISMA checklist. When completing the checklist, please use section and paragraph numbers, rather than page numbers.

a) Please add the following statement, or similar, to the Methods: "This study is reported as per the Preferred Reporting Items for Systematic Reviews and Meta-Analyses (PRISMA) guideline (S1 Checklist)."

7) In line with PLOS Medicine’s guidelines, please update your search to the present time.

8) In figures 2, 3, 4, please show the axis beginning at zero for both graphs. If this is not possible, please show a break in the axis.

9) Please define the abbreviations in Tables and Figures e.g., LIC, LMIC, UMIC

10) Please confirm that the appropriate usage rights apply to the use of this map. Please see our guidelines for map images: https://journals.plos.org/plosmedicine/s/figures#loc-maps

11) References: 

a) Please ensure that journal name abbreviations consistently match those found in the National Center for Biotechnology Information (NCBI) databases. https://journals.plos.org/plosmedicine/s/submission-guidelines#loc-references. 

12) Please remove the ‘Data sharing” and “Declaration of interests” from the end of the main text. In the event of 

publication, this information will be published as metadata based on your responses to the submission form.

Comments from the reviewers:

Reviewer #1: I confine my remarks to statistical aspects of this paper. The general approach is fine, but I have some issues to reolve before I can recommend publication.

The lack of line numbers made the review process harder.

The general approach is OK, but I have some issues to resolve before I can recommend publication.

On p. 2, you give the mean age with a +- figure, but you don't say what this is (SD? Standard error? Something else?) The other numbers are given with 95% CI.

p. 5 When you say things like "leading cause" please also give rates.

p. 9 I don't fully understand the first sentence in data analysis about dividing it two. Why was this done? Maybe I am missing something.

Peter Flom

Reviewer #2: The authors provide a review report documenting the urban-rural difference in hypertension in low- and middle-income countries. The strength of this paper is that this is the first systematic review focused on urban-rural differences in the prevalence of hypertension in LMICs. The results highlighted the importance of hypertension prevention and control even in rural areas. There is significant heterogeneity in the methodology of the included studies. There is no discussion about risk factors, awareness, and control of hypertension, which could make this review more beneficial and valuable. My comments are as follows:

Major comments:

1) Time trend analysis using cross-sectional surveys in different populations would have several limitations. Calculating the rate of change using two different populations conducted in different areas may be misleading. Meta-regression may not be the best method for this type of data. 

2) Different studies are using different definitions of rural and urban areas. Would it be possible to do a sensitivity analysis by including studies that used the consistent definition? 

3) It would be great if authors would also have data on risk factors. Why authors did not extract them? A reasonable explanation for rural-urban variation is incomplete. 

Minor comments:

5) Presenting the prevalence of hypertension in terms of range would be easier for many readers. 

6) The authors presented results for HDI and IMR but the detailed methodology is missing. Which year HDI and IMR were used? Did the author account for data collection year or paper published year? 

7) What do you mean by representing 87% of LMICs population by 2018?

8) As the authors had discarded non-English papers it would be interesting to know how much data was missed.

9) What was the response rate for each study? Did the studies use a random sampling technique for sample selection? It would also be interesting to know whether data were collected at home or hospital setting or elsewhere. 

10) Self-reported data on the diagnosis of hypertension is interesting in the context of LMICs. Would it be possible for authors to run a sensitivity analysis by excluding those who self-reported the diagnosis of hypertension? 

Reviewer #3: This is an interesting review highlighting urban-rural variation in hypertension prevalence in low- and middle-income countries (LMICs). Especially given the ongoing double and or triple burden of diseases in LMICs and the rapid urbanisation taking place in these settings.

The introduction, data analysis, results, and discussion sections were well written and I noted the comparisons with previous regional reviews.

However, I will strongly recommend that the authors update their search as they conducted their search from 01/01/1990 to 01/05/2018, their search is almost four years old and although publications in the last two years might have been dominated by the COVID-19 pandemic, I suggest that the search be updated to at least 31/12/2021.

Otherwise, this study has the potential to influence public health decision making and resource allocation towards interventions targeting prevention and management of hypertension in LMICs.

Reviewer #4: This is a well-written manuscript. I have a few general comments:

1. The author may need to include non-English articles as a lot of low-and middle-income countries (China, Iran, Vietnam)are non-English speaking countries.

2. In the method section, I would suggest justifying the reason for including the country-level socio-economic variables in the regression model. 

3. Figure 3 indicates the large heterogeneity in HDI and other indicators among countries our different income groups. Does the control of the region, income status (LIC, LMIC, UMIC,) and country-level socio-economic are controlled in the regression lead to overfitting?

4. Given the heterogeneity in Urban-rural differences in hypertension across countries and regions, the author adjust the fixed effect of countries and/or regions? have the author considered the three-level meta-regression?

[LINK]

---

## [Decision Letter · Decision Letter 2]

19 Jul 2022

Dear Dr. Ranzani,

Thank you very much for re-submitting your manuscript "Urban-rural differences in hypertension prevalence in low-income and middle-income countries, 1990-2020: a systematic review and meta-analysis" (PMEDICINE-D-21-03953R2) for review by PLOS Medicine.

I have discussed the paper with my colleagues and the academic editor and it was also seen again by three reviewers. I am pleased to say that provided the remaining editorial and production issues are dealt with we are planning to accept the paper for publication in the journal.

[LINK]

We look forward to receiving the revised manuscript by Jul 26 2022 11:59PM.   

Sincerely,

Beryne Odeny, 

PLOS Medicine

plosmedicine.org

Requests from Editors:

1) Abstract: Methods and findings – please add inclusion and exclusion criteria for studies

2) Abstract: please move the following statement "This systematic review and meta-analysis has been registered with PROSPERO (CRD42018091671) from the "Conclusions" to the "Methods and findings" section.

Comments from Reviewers:

Reviewer #1: The authors have addressed my concerns and I now recommend publication

Peter Flom

Reviewer #2: The authors have adequately addressed and/or justified for my comments. 

Reviewer #4: Thank you for improving the article. The author has sufficiently addressed reviewers' questions.

[LINK]

---

## [Editor Report · Decision Letter 3]

22 Jul 2022

Dear Dr Ranzani, 

On behalf of my colleagues and the Academic Editor, Dr. Sanjay Basu, I am pleased to inform you that we have agreed to publish your manuscript "Urban-rural differences in hypertension prevalence in low-income and middle-income countries, 1990-2020: a systematic review and meta-analysis" (PMEDICINE-D-21-03953R3) in PLOS Medicine.

PRESS

Sincerely, 

Beryne Odeny 

PLOS Medicine